# Updated Prognostic Factors in Localized NSCLC

**DOI:** 10.3390/cancers14061400

**Published:** 2022-03-09

**Authors:** Simon Garinet, Pascal Wang, Audrey Mansuet-Lupo, Ludovic Fournel, Marie Wislez, Hélène Blons

**Affiliations:** 1Pharmacogenomics and Molecular Oncology Unit, Biochemistry Department, Assistance Publique—Hopitaux de Paris, Hôpital Européen Georges Pompidou, 75015 Paris, France; simon.garinet@aphp.fr; 2Centre de Recherche des Cordeliers, INSERM UMRS-1138, Sorbonne Université, Université de Paris, 75006 Paris, France; 3Oncology Thoracic Unit, Pulmonology Department, Assistance Publique—Hopitaux de Paris, Hôpital Cochin, 75014 Paris, France; pascal.wang@aphp.fr (P.W.); marie.wislez@aphp.fr (M.W.); 4Pathology Department, Assistance Publique—Hopitaux de Paris, Hôpital Cochin, 75014 Paris, France; audrey.lupo@aphp.fr; 5Thoracic Surgery Department, Assistance Publique—Hopitaux de Paris, Hôpital Cochin, 75014 Paris, France; ludovic.fournel@aphp.fr

**Keywords:** resected NSCLC, prognosis, adjuvant chemotherapy

## Abstract

**Simple Summary:**

In resected lung cancer, adjuvant and neoadjuvant chemotherapy following surgery are currently mainly based on TNM classification. With the validation and ongoing trials of targeted therapies in this situation, the relapse risk evaluation needs to be improved to better discriminate patients who will really benefit from adjuvant therapies. The objective of this review is to put forth an update on the identified clinical, pathological and molecular prognostic factors and biomarkers in development that could change clinical practices in the near future.

**Abstract:**

Lung cancer is the most common cause of cancer mortality worldwide, and non-small cell lung cancer (NSCLC) represents 80% of lung cancer subtypes. Patients with localized non-small cell lung cancer may be considered for upfront surgical treatment. However, the overall 5-year survival rate is 59%. To improve survival, adjuvant chemotherapy (ACT) was largely explored and showed an overall benefit of survival at 5 years < 7%. The evaluation of recurrence risk and subsequent need for ACT is only based on tumor stage (TNM classification); however, more than 25% of patients with stage IA/B tumors will relapse. Recently, adjuvant targeted therapy has been approved for EGFR-mutated resected NSCLC and trials are evaluating other targeted therapies and immunotherapies in adjuvant settings. Costs, treatment duration, emergence of resistant clones and side effects stress the need for a better selection of patients. The identification and validation of prognostic and theranostic markers to better stratify patients who could benefit from adjuvant therapies are needed. In this review, we report current validated clinical, pathological and molecular prognosis biomarkers that influence outcome in resected NSCLC, and we also describe molecular biomarkers under evaluation that could be available in daily practice to drive ACT in resected NSCLC.

## 1. Introduction

Based on GLOBOCAN estimates of lung cancer incidence and mortality in 2020, lung cancer represented 11.4% of the 19.3 million cases and remained the leading cause of cancer death with 1.8 million deaths [1]. It is classified into non-small cell lung cancer (NSCLC), which accounts for 80–85% of cases, and small cell lung cancer (SCLC). Although tobacco is the major risk factor for lung cancer, 10–15% of patients in Caucasians and up to 40% in Asians are non-smokers. Risk factors and disease etiology remain largely unknown in non-smokers. NSCLC is not usually diagnosed until advanced-stage disease is present [2].

Analysis of stage distribution at diagnosis in the 2014–2018 period showed that approximately ¼, ¼ and ½ of patients had localized, regional and distant stage disease. Indeed, patients with metastatic cancer at diagnosis still represent 46% of patients, although the proportion of disease diagnosed at a localized stage increased from 17% during the mid-2000s to 28% in 2018, thanks to cancer screening and prevention campaigns [3].

Carcinogenesis is linked to the presence of somatic molecular alterations in specific oncogenic drivers, some of which are druggable. Alterations in the RAS/MAP kinases pathway led to the development of specific drugs, targeting EGFR mutations, MET exon14 skipping, ALK/RET/ROS1/NTRKs fusions and recently the KRAS G12C mutation hotspot; therefore, lung adenocarcinoma is the solid tumor with the highest number of validated targeted therapies, and thus it is a genomic model of mutation-driven cancer therapy for metastatic disease [4]. Moreover, ICI showed good results for NSCLC, especially in the subgroups of patients without driver alteration, and is now approved in first-line therapy in combination with platinum-based chemotherapy for treatment of stage IV disease. However, except in rare cases of patients with complete long-term pathological responses, there is no curative objective for metastatic disease. On the other hand, the treatment goal for patients with early-stage lung cancer is cure with a complete surgical resection [5]. However, 30% to 50% of patients will relapse even after an R0 complete resection. Consensus guidelines support adjuvant or neoadjuvant chemotherapy for tumors with higher risk of recurrence [6]. Risk stratification is thus crucial to identify patients who will really benefit from adjuvant treatment. However, few prognostic factors are validated for localized NSCLC, and this stratification is actually only based on stage determination in pathology. This strategy implies that many patients undergo chemotherapy, while they do not obtain any benefit from it, as they would not have relapsed. These patients are unnecessarily exposed to cisplatin-based chemotherapy adverse effects such as nephrotoxicity but also anaphylaxis, cytopenia, hepatotoxicity, ototoxicity, cardiotoxicity, nausea and vomiting, diarrhea, mucositis, stomatitis, pain, alopecia, anorexia, cachexia and asthenia. Moreover, genomic-alteration-driven adjuvant chemotherapy, a paradigm derived from the good results for metastatic diseases, is currently a hot topic in localized NSCLC. ADAURA Trial concluded with a significantly longer DFS at 24 months for EGFR-mutated patients treated with osimertinib: 89% (95% CI, 85 to 92), versus the placebo group 52% (95% CI, 46 to 58) [7]. The Adjuvant Lung Cancer Enrichment Marker Identification and Sequencing Trials (ALCHEMIST) is a group of randomized clinical trials for patients with early-stage NSCLC whose tumors have been completely removed by surgery. ALCHEMIST-EGFR with erlotinib versus placebo for patients with EGFR mutations and ALCHEMIST-ALK for crizotinib versus placebo for patients with ALK translocation are ongoing. Adjuvant treatment protocols with targeted therapies could be of long duration, lasting 3 years for EGFR TKI. Finally, ALCHEMIST-immunotherapy is ongoing (NCT02595944) for the evaluation of one-year nivolumab therapy after surgery and adjuvant chemotherapy for patients with stage IB-IIIA disease versus observation only. Another trial is ongoing (NCT03254004) for adjuvant pembrolizumab in stage I resected lung adenocarcinoma. Moreover, the role of immunotherapy in the neoadjuvant setting is being analyzed using the pathological response as primary endpoint in most cases. This situation stresses the need for better prognosis stratification in localized NSCLC in order to identify patients with the highest risk of relapse and avoid unnecessary treatment for others. In this review, we report current clinical, pathological and molecular markers used for prognosis assessment in localized NSCLC and discuss future prognostic biomarkers.

## 2. Clinical Prognostic Factors Related to the Patient

### 2.1. Age, Gender and Performance Status

Age represents an important independent pejorative prognostic factor, especially when it contraindicates the surgical act [8,9]. Indeed, several articles showed that the older the person, the greater was the risk of early death from localized lung cancer (Table 1) [10,11,12]. In contrast, relapse-free survival after surgical resection is independent of the patient’s age and an elderly person (>70 years old) in good general condition will therefore benefit as much from surgery at this point as a younger subject (Table 1) [13].

Whatever the stage of the disease, gender is a prognostic factor. Indeed, thoracic surgeons and oncologists tend to agree that women operated on for non-small cell lung cancer have a better prognosis at the same age and stage than men [14,15]. Indeed, last year, Sachs et al. confirmed in a nationwide cohort of 6356 patients that women who were operated on for lung cancer had a significantly better prognosis than men (lower risk of death, HR = 0.73; 95% CI (0.67–0.79) and this survival advantage was found, regardless of age, common comorbidities, physical performance, tumor characteristics and stage of disease [15]. Interestingly, the study by Cerfolio et al. even found that women who received neoadjuvant chemotherapy (*n* = 76) had a more partial or complete response than men (*n* = 142) (ORRwoman = 79% vs. ORRman = 51%; *p* = 0.025) [14]. The reasons for such a difference are still unknown, but a few articles suggest a likely relation with estrogen and progesterone receptors [16,17].

The prognostic value of “performance status (PS)” remains debated. Indeed, this scale evaluates the patient’s capacity to carry out daily activities [8,18], dividing it into six levels that will help in choosing a therapeutic plan (Table 2) [18,19]. Several studies demonstrated a straight correlation between PS and prognosis and showed that PS is a clinical parameter with high confidence in metastatic settings. In localized NSCLC, a Japanese study by Kawaguchi et al. (*n* = 26,957 patients) showed this in stage I AJCC patients (*n* = 9333) that had a PS of 0, 2, 3 and 4 in 7179 (76.9%), 247 (2.6%), 86 (0.9%) and 21 (0.2%) patients, respectively. Median overall survival was dramatically decreased between groups with PS 0 (91.6 months), PS 3 (30.9 months) and PS 4 (10.0 months) (*p* < 0.0001) [20]. Moreover, Powell et al. showed that a poor PS at diagnosis was correlated to a high risk of early death after resection (within 90 days of surgery) (PS = 0 vs. PS > 2: OR 4.08, 95% (2.37–7.02)) (Table 1) [10].

### 2.2. Nutritional and Morphometric Parameters

Recently, nutritional status at diagnosis has been increasingly studied, especially in localized stages. Several studies showed that overweight and obesity were associated with lower lung cancer incidence, whereas in other cancers, such as gallbladder and kidney cancers, the risk of cancer is linearly associated with each 5 kg/m^2^ increase [21]. Some authors even talk of a “lung cancer paradox” to describe the observation of a protective effect of BMI on lung cancer [22]. Moreover, Alifano et al. argued that the “lung cancer paradox” is also true for overall survival and showed that preoperative BMI is a strong and independent predictor of survival in patients undergoing surgery for resectable lung cancer. This large-scale study included more than 54,000 patients who were divided into four groups (underweight (BMI < 18.5 kg/m^2^), normal weight (18.5 ≤ BMI < 25 kg/m^2^), overweight (25 ≤ BMI < 30 kg/m^2^) and obese (30 ≥ kg/m^2^)) and each abnormal group was compared to the normal weight. After adjusting for the period of study, age, sex, WHO performance status, comorbidities, side of tumor, extent of resection, histologic type and stage of disease, Alifano et al. found that underweight was associated with lower survival (HRs 1.51, 95% (1.41–1.63)) compared to normal weight, whereas overweight and obesity were associated with improved survival (0.84, 95% (0.81–0.87) and 0.80, 95% (0.76–0.84), respectively) [23]. This large French study confirms the recent concept of “obesity paradox” in resectable lung cancer patients [24]. In addition, to understand this paradox, his team also found recently that low muscle mass or sarcopenia (defined as a value below the 33 percentiles of Index Total Muscular Mass (iTMM) of sex-specific population) before surgery has long-term negative impact on non-small cell lung cancer (NSCLC) (seven years survival rates were 31.6% and 50.1% in the presence of low and normal muscle mass, *p* = 0.042, respectively) and, on the contrary, the absence of low muscle mass was an independent favorable prognostic (RR = 0.56 (0.37–0.87), *p* = 0.00091). Interestingly, in this study, low muscle mass was less common in obese (9.5%) than in underweight patients (66%), which may partly explain the “obesity paradox” [25]. Hence, strategies improving body fat and muscular mass before surgery are largely recommended in localized stages.

In accordance, a Danish study by Christensen et al. involving stage I lung cancer showed that low nutritional status at diagnosis (BMI < 18.5 kg/m^2^) was associated with a higher risk of early death compared to normal nutritional status (18.5 ≤ BMI ≤ 24.9 kg/m^2^) (OR 2.3; 95% CI (1.4–3.7)) [26]. Hence, low nutritional status seems to be a determining factor of poor prognosis in localized lung cancers.

### 2.3. Smoking and Alcohol Exposure

Active smokers at diagnosis have a worse prognosis and an increased risk of death of one-third as compared to former or non-smokers [26,27,28]. In addition to its carcinogenic effect, tobacco causes multiple comorbidities such as chronic obstructive pulmonary disease (COPD) or cardiovascular complications which directly impact the postoperative prognosis [29,30,31,32]. Regarding chronic alcoholism, patients with stage I NSCLC and high-risk alcohol consumption (defined by more than three drinks per day in men and two drinks per day in women) have a higher risk of premature death than nonalcoholics (OR = 2.2, 95% CI (1.4–3.5)) [32]. It is therefore important to encourage patients to quit tobacco and alcohol before surgery [28]. In a recent prospective study by Sheikh et al., it was demonstrated that smoking cessation after diagnosis improved overall survival (mOSquit = 6.6 years vs. mOSno-quit = 4.8 years, *p* = 0.001) and progression-free survival (PFS 5-year quit = 54.4% vs. PFS 5-year no-quit= 43.8%, *p* = 0.004). These effects were observed even among mild to moderate smokers [27].

### 2.4. Other Comorbidities and Symptoms at Diagnosis

Literature is scarce but comorbidities and the presence of symptoms at diagnosis are potential clinical prognostic factors.

The presence of comorbidities at diagnosis assessed by the Charlson comorbidity index seems to play a significant role in the localized stages. The prognostic value is even higher in patients with another primary cancer [33,34]. Indeed, in studies by Powell et al. and Christensen et al., patients who died early had more comorbidities, and 25% of early deaths were related to comorbidities. The main cause in this group was death from another primary malignant disease (e.g., digestive cancer, renal cancer and melanoma) [10,34].

Recently, an American retrospective study on 3045 early-stage resected NSCLC found that patients with common comorbidities such as COPD (HR = 1.25; 95% CI (1.06–1.49)), coronary artery disease (CAD) (HR 1.17, 95% CI (0.99–1.39)) and diabetes (HR = 1.20; 95% CI (1.0005–1.43)) had poor OS [12].

Interestingly, an Italian prospective study in stage I NSCLC showed that the presence of symptoms and particularly systemic symptoms (fever, weight loss, asthenia) had an unfavorable impact on progression-free survival and overall survival (RRDFS = 2.0, 95% CI (1.2–3.4); *p* = 0.003 and RRsurvival = 2.2, 95% CI 95% (1.3–3.8); *p* = 0.003) compared to the presence of local respiratory symptoms (cough, chest pain, dyspnea) [35]. Taken together, systemic inflammation evaluated by preoperative CRP level and nutritional status may predict outcome [36,37].

**Table 1 cancers-14-01400-t001:** Main clinical characteristics associated with early lung cancer death in resected early-stage NSCLC [38].

Reference	Number	Age (Years)	Male Sex	Tobacco	ECOG PS	CCI (vs. 0)	Nutritional Status
Powell et al. [8]	10,991	70–74 vs. >85: OR 2.84 (1.71;4.71)	OR 1.37 (1.15;1.63)	NA	0 vs. 1: OR 1.38 (1.09;1.75) 0 vs. 2: OR 2.40 (1.68;3.41) 0 vs. >2: OR 4.08 (2.37;7.02)	2–3: OR 1.54 (1.25;1.90) ≥4: OR 1.53 (1.07;2.18)	NA
Stoelben et al. [9]	1281	≥75: RR 2.46 (1.17;5.16)	RR 1.51	NA	NA	NA	NA
Currow et al. [39]	304	<60 vs. 60–69: HR 1.25 (1.05;1.49) 70–79: HR 1.46 (1.23;1.73) ≥80: HR:1.86 (1.54;2.24)	NS	NA	NA	NA	NA
Melvan et al. [30]	215,645	NA	OR 1.55 (1.44–1.65)	NA	NA	1: OR 1.12 (1.04;1.20) ≥2: OR 1.56 (1.43;1.70)	NA
Christensen et al. [24]	2985	NA	NA	Non-smoker: OR 0.3 (0.1;0.9)	NA	NA	BMI < 18.5 kg/m^2^ vs. 18.5 ≤ BMI≤ 24.9 kg/m^2^: OR 2.3; 95% CI (1.4–3.7)
Friedel et al. [29]	595	NA	NA	≥40 pack-years: HR 1.40 (1.05;1.86)	NA	NA	NA

BMI = body mass index, ECOG PS = Eastern Cooperative Oncology Group performance status; CCI = Charlson comorbidity index; OR = odds ratio; HR = hazard ratio; RR = relative risk; NA = not available; NS = non-significant.

## 3. Histopathological Prognostic Factors Related to the Tumor

At diagnosis, only a third of patients present an operable NSCLC. After complete resection, there is great heterogeneity of prognosis that makes necessary the identification of relevant prognostic factors which could help the physician to propose the most suitable therapeutic management [13].

### 3.1. The TNM Classification

The TNM classification was established for this purpose and has been constantly improved since its first publication in 1968. Indeed, many changes have been made since its creation and these changes have led to the eighth edition. This latest edition is based on the analysis of survival data of more than 94,700 patients with NSCLC stage I to IV, treated in different countries around the world [40].

Staging of NSCLC is the most objective and reproducible prognostic factor [8,9,40]. It considers the size of the tumor (T), the locoregional lymph node invasion (N) and the metastasis status (M). The clinical TNM (cTNM) is assessed by radiological imaging while the pathological TNM (pTNM) is based on the anatomo-pathological analysis of the resected tumor and is considered to be the most reliable.

#### 3.1.1. Size (T)

In stage I operated patients, tumor size is one of the most important prognostic factors. Already in the 2000s, in a retrospective analysis of more than 19,000 cases of stage I, the authors had demonstrated its importance. Indeed, patients with a tumor measuring less than 4 cm had a five-year survival rate of 48.8% versus 35.5% for patients with a tumor of more than 4 cm [41]. The eighth classification confirms the statement that every centimeter counts and continues to further divide T1 and T2 groups into subgroups according to the tumor size: T1a ≤ 1 cm, 1 cm < T1b ≤ 2 cm, 2 < T1c ≤ 3 cm, 3 cm < T2a ≤ 4 cm, 4 < T2b ≤ 5 cm, T3 between 5 and ≤7 cm and T4 > 7 cm. The risk of death increases with the size of the tumor: if the hazard ratio is 1 for pT1aN0, it will be 1.55 for pT1b2N0, 2.07 for T1cN0, 2.83 for pT2aN0 and 3.89 for pT2bN0 [40].

#### 3.1.2. Nodes (N)

Although no change has been made to the classification of lymph node extension since the seventh edition, it remains a major negative prognostic factor. Indeed, Gajra et al. demonstrated that the number of nodes examined is directly correlated with recurrence-free survival. Moreover, lymph status can only be properly established if more than six nodes are removed and studied [42]. Indeed, in this study, the five-year recurrence-free survival of N0 patients with fewer than six lymph nodes examined was 52%, while for those with more than six lymph nodes studied it was 75%. Among N1 patients, those with only intrapulmonary lymph nodes have a prognosis approaching N0 patients, better than patients with invaded hilar lymphadenopathy [43]. Within the N2′s situations, two scenarios seem to be more favorable: N2 involvement without N1 involvement (“skipping metastases”) and N2 involvement localized to a single lymph node [44,45]. Hence the seventh TNM classification defines as follows: invasion of a single station N1 (N1a), multiple stations (N1b), single station N2 without reaching N1 (skipping metastases: N2a1), single station N2 and reaching N1 (N2a2) and invasion of multiple stations N2 (N2b) [46]. Finally, N3 disease classifies patients as stage IIIB and generally contraindicates surgical excision.

#### 3.1.3. Metastatic Invasion (M)

Patients with metastatic disease are not candidates for surgery; however, it is possible to accidentally discover, during surgery or during anatomo-pathological examination, a second anatomo-pathological nodule or a pleural invasion, which darkens the prognosis of patients. Since the seventh TNM, a nodule in the same lobe classifies the tumor as pT3, a nodule in another lobe but in the same lung classifies the tumor as pT4, while a nodule in the contralateral lung is a pM1a metastasis, with a survival rate at five years of 53%, 36% and 10%, respectively [40].

#### 3.1.4. Future Perspectives with the Ninth TNM Staging

A new ninth edition of the TNM classification is scheduled for publication in 2024 and some novelties will be potentially introduced. There are novelties concerning the classic “anatomical” parameters of TNM. An example of this is a more accurate description of the invasion of the chest wall with the involvement of its different layers (parietal pleura, bony structures and soft tissue) which could potentially show a different prognosis [47]. Indeed, Sakakura et al. found that an operated patient with T3 tumors with invasion of the parietal pleura alone had better survival that a patient with deeper involvement of the chest wall (5-year survival rates 50% and 36.7%, respectively, *p* = 0.028). Hence, there is a proposal to subdivide the eighth edition T3 category into T3a (invasion of parietal pleura only) and T3b (invasion of deeper structures of the chest wall) [48]. In addition, the ninth edition will confirm whether only the invasive part of the lepidic adenocarcinoma should be considered instead of the total size for prognosis. Improvements will also concern the interpretation of the N status integrating the knowledge of the nodal status post induction therapy. This will delineate the prognostic impact of the N classification in patients receiving induction therapy prior to surgery.

### 3.2. Histological Type

The histological type is classically associated as an independent prognostic factor, without being perfectly reproducible [8]. Concerning the two most common histologies according to the WHO 2004 criteria, survival rate in resected stage I seems higher for squamous cell carcinoma than for adenocarcinoma and approximately 80% of patients with squamous cell carcinoma are alive five years after diagnosis compared to approximately 70% of similarly staged adenocarcinoma [49]. In addition, several surgical studies suggest that in resectable diseases, patients with squamous cell carcinoma have a better prognosis than patients with adenocarcinoma after adjustment on stage [9,20]. However, the prognostic value of each histology type is still being debated due to the continuously adapting, specific terminology and criteria used to distinguish squamous cell carcinoma from adenocarcinoma, particularly in poorly differentiated tumors [50,51]. The major changes within the new WHO 2021 classification are as follows: lymphoepithelial carcinoma is now part of squamous cell carcinoma; the classification of lung neuroendocrine neoplasm has been updated on evolving concepts classification; the recognition of bronchiolar adenoma/ciliated muconodular papillary tumor (BA/CMPT) as a new entity within the adenoma subgroup and the recognition of thoracic SMARCA4-deficient undifferentiated tumor [52].

Concerning less common histologic types, pulmonary sarcomatoid carcinomas and SMARCA4-deficient carcinoma with its dedifferentiated counterpart, the SMARCA4-deficient undifferentiated thoracic tumor (SMARCA4-UT), are significantly associated with worse prognosis. For the latter, it is now recognized as an entity in the WHO 2021 classification of lung tumors [52].

Pulmonary sarcomatoid carcinomas are a rare group of tumors accounting for about one percent of NSCLC. Patients are usually men over 60 years old, tobacco smokers and frequently symptomatic (80%). Tumors are voluminous, more often peripherical than central, more aggressive, with strong fixation on 18 F-fluorodeoxyglucose positron-emission tomography (18FDG/TEP) [53,54]. SMARCA4-UT is characterized by inactivating mutations of SMARCA4, a suppressor tumor gene, resulting in loss of expression of brahma-related gene (BRG1). These tumors are often large at presentation with a massive invasion of the anterior mediastinum, pulmonary hilum and sometimes with invasion of the chest wall. Like pulmonary sarcomatoid carcinomas, patients are usually male and heavy smokers, but they are younger (40–50 years old) [55,56,57].

Both subgroups have a poor prognosis as compared to other NSCLC subtypes because of greater aggressiveness and frequent chemoresistance. As the sarcomatoid subtype is a recognized prognostic marker, some teams recommend adding adjuvant chemotherapy in resected pulmonary sarcomatoid carcinoma independently of disease staging [58].

### 3.3. Histological Subtype of Adenocarcinoma and Grade

In 2016, a histo-prognostic classification based on architectural predominance of adenocarcinoma determined on surgical resection was introduced by the International Association for the Study of Lung Cancer (IASLC), American Thoracic Society (ATS) and European Respiratory Society (ERS) (Table 2). This classification allows the possibility to select patients associated with a poor prognosis (micropapillary, solid, colloid and invasive mucinous) in order to treat them with a potential adjuvant therapy [59,60]. In addition, in the acinar-predominant pattern group, Mansuet-Lupo et al. showed that the determination of cribriform pattern allowed a better identification of patients with poorer survival. Indeed, the five-year survival rate for patients with acinar adenocarcinomas containing > 10% cribriform areas (51.3%) was intermediate between the remaining acinar-predominant adenocarcinomas (58.2%) and the solid-predominant adenocarcinoma (45.1%) [61].

**Table 2 cancers-14-01400-t002:** Disease-free survival at 5 years, according to IASLC/ATS/ERS adenocarcinoma histological subtypes (adapted from [60]).

IASLC/ATS/ERS Classification Subtypes	Number (%)	Disease Free Survival 5 Years
**Low grade**		
In situ adenocarcinoma	1 (0.2%)	100%
Minimally invasive adenocarcinoma	8 (1.2%)	100%
**Intermediate grade**		
Lepidic predominant	29 (6%)	90%
Acinar predominant	232 (45%)	84%
Papillary predominant	143 (28%)	83%
**High grade**		
Micropapillary predominant	12 (2%)	67%
Solid predominant	67 (13%)	70%
Colloid predominant	9(2%)	71%
Invasive mucinous adenocarcinoma and invasive mixed (mucinous/non-mucinous) adenocarcinoma	13 (3%)	76%

A few years later, a new study from the IASLC’s group demonstrated that the combination of the predominant histological pattern associated with the pattern of the worst prognosis was more efficient and better predicted the evolution than the classification according to the histological subtype alone. Therefore, in 2020, the IASLC proposed a new classification by grading. This grading proposes to consider the predominant architectural pattern and the presence or absence of foci of solid architecture or micropapillary or cribriform or complex glandular if it represents more than 20%. Complex architecture represents areas with fused irregular glands or single cell infiltrating a desmoplastic stroma [62].

### 3.4. Pleural and Lymphovascular Invasion

As evidenced by the eighth TNM version, pleural per-contiguity invasion represents an established independent pathological feature associated with poor prognosis [40,63]. In fact, NSCLC tumors with pleural infiltration exhibit stronger invasive potential, with a higher risk of pleural cavity dissemination and mediastinal lymph node spread. Pleural invasion is classified into the following subgroups: PL0 in the absence of elastic layer invasion; PL1 if the tumor invades the elastic layer; PL2 in case of pleural surface invasion; and PL3 if cancer invades the parietal pleural invasion [64,65]. A recent meta-analysis of 16 studies documented those patients with PL2 had poorer overall survival and five-year survival than those with PL1 [66].

The presence of lymphatic or vascular emboli is defined as the embolization of tumor cells in the vascular lumen or lymphatic ducts. Its impact on the survival of patients with surgically resected stage I NSCLC is quite controversial [67]. However, a French study and two meta-analyses support the proposal that lymphovascular invasion is a poor prognostic factor in terms of relapse and death [67,68,69].

### 3.5. Spread through Air Spaces (STAS)

Another important prognostic factor is STAS which is now recognized in the new 2021 WHO Classification of Lung Tumors as a histological feature with prognostic significance [52]. It is defined as the presence of tumor cells within the air spaces in the lung parenchyma, beyond the edge of the main tumor. Kadota et al. showed that stage I lung adenocarcinomas with STAS, treated with limited resection, have a higher risk of distance and locoregional recurrence. Indeed, “STAS” is an independent significant risk factor of disease recurrence in early stages [70,71], with a five-year cumulative incidence of recurrence statistically higher in STAS-positive (42.6%) than in STAS-negative (10.9%) (*p* < 0.001) [70]. Since then, other studies have shown the same poor prognostic significance of STAS and even for other histological types such as squamous type [72].

## 4. Tumor Molecular Alterations

Comprehensive molecular profiling has shown a high degree of molecular heterogeneity in lung cancer. Molecular alterations are used in clinical practice to select treatment for lung cancer patients. Studies have revealed associations between molecular alterations in oncogene drivers and response to targeted therapies; however, links with prognosis remain a matter of debate. For patients with metastatic disease, the outcome largely depends on the identification of a targetable driver such as EGFR, ALK, ROS1, ERBB2, BRAF, MET and more recently KRAS, RET, NRG1 and NTRK1-2-3, as mutations or gene fusions are highly predictive of response to matched targeted therapy. For patients with resectable tumors, systematic molecular profiling in care settings is about to start since adjuvant targeted therapies have been or are being tested [7]. Because none of these treatments are completely harmless, and because adjuvant treatments may last for years, it is important to determine the benefit/risk balance of adjuvant therapy. The identification of molecular prognostic markers could help select patients for adjuvant targeted therapies.

### 4.1. Molecular Drivers

#### 4.1.1. EGFR

Mutations in the epidermal growth factor receptor (EGFR) commonly occur in NSCLC, ranging from 13% in Caucasians to 44% in East Asians. Discordant data exist in the literature concerning the prognostic value of EGFR mutation in resected lung cancer, showing both positive and negative prognostic value [73,74,75]. A recent meta-analysis of 19 studies involving 2086 EGFR mutated among 4872 localized NSCLC concluded that DFS of EGFR-mutated patients was similar to wild-type patients [76]. They found a trend for a slightly lower DFS in patients with EGFR DEL19 mutated tumors as compared to L858R tumors (HR 1.38, 95% CI: 0.76 to 2.52). However, EGFR mutations have a strong predictive value of response to TKI-EGFR both in metastatic and adjuvant settings [7,77] (ADAURA). Masago et al. explored a group of patients (17/512) with long-term recurrence after complete resection for early-stage lung cancer and showed that all but one patient with late recurrences had driver mutations, including 11/17 with EGFR mutations [78]. EGFR-mutated localized NSCLC could have a higher risk of long-time relapse. This could be explained by the higher ability of EGFR mutant cells to form distant early micrometastases [79], or by the observation that driver-mutation–related tumors have low tumor mutational burden and are more likely to escape immune surveillance [80].

#### 4.1.2. KRAS/BRAF

Most studies identify no significantly prognostic value for KRAS mutation status in localized NSCLC [81]. However, in subgroup analyses, KRAS could have a bad prognostic impact on stage I tumors. Adjuvant CT based on KRAS testing is not recommended [82].

Patients with BRAF-mutated tumors represent a highly aggressive subtype of colon cancer, both in metastatic and localized situations [83]. In resected NSCLC, the overall survival rate was not significantly different between patients with wild-type BRAF and those with V600E or non-V600E BRAF mutations [84].

#### 4.1.3. ALK

Liu et al. showed in a cohort of 2103 resected patients in which 81 were ALK-positive that ALK rearrangement was not an independent prognostic factor in stage I–IIIA lung adenocarcinoma [85]. Fukui et al. selected adenocarcinoma cases who underwent pulmonary resection and reported a five-year OS rate of 81% and 77% (*p* = 0.76) for ALK-positive and ALK-negative, respectively [86]. Concerning DFS, Paik et al. reported a median DFS of 76.4 months in ALK-positive and 71.3 months in ALK-negative (EGFR status unknown) cases (*p* = 0.66) in resected stage I–III NSCLC patients. However, others showed controversial results. In stage IIIA, ALK-positive patients had poorer DFS than ALK-negative patients (median DFS, 6 months versus 16 months, *p* = 0.0057, Table 3). In a multivariate analysis, the ALK-positivity was the only significant variable associated with poor survival in stage IIIA NSCLC (HR = 4.0, *p* < 0.001) [87].

#### 4.1.4. MET

MET exon 14 skipping mutations occur for 1–2% of lung adenocarcinomas. This alteration was not found to have a prognostic impact in localized disease [88]. MET inhibitors are available in metastatic situations for MET ex14 mutated or MET amplified [89]; however, they are not evaluated in ACT and their predictive value remains unknown in localized NSCLC.

### 4.2. Tumor Suppressor Genes

#### TP53/STK11/KEAP1

The prognostic value of mutations in the tumor suppressor gene TP53 has been widely evaluated in retrospective studies [81,90,91,92] but has not been validated due to controversies in the literature. The wide heterogeneity of TP53 mutations on P53 protein residual function may in part explain differences in results. The presence of a TP53 mutation does not necessarily imply complete P53 inactivation; mutations are classified from total loss-of-function to gain-of-function mutations. A meta-analysis on the prognostic impact of TP53 mutations concludes that TP53 mutations lead to shorter OS in localized NSCLC, stage I–IIIA [93]. Saleh et al. confirmed in a large cohort of 1518 surgically treated patients that truncating TP53 mutations and KEAP1 mutations were independent negative prognostic markers in multivariable analysis (hazard ratio [HR] TP53 truncating = 1.43, 95% confidence interval [CI]: 1.07–1.91, *p* = 0.015; HR KEAP1mut = 1.68, 95% CI:1.24–2.26, *p* = 0.001) with shorter OS and DFS [94]. The prognostic value of KEAP1 mutations was more recently assessed. KEAP1 inactivation leads to a derepression of NRF2 and consequently improved oxidative stress responses and growth advantage.

STK11/LKB1 is a tumor suppressor commonly mutated in lung cancer and involved in the mTOR pathway. Prognostic value has been widely evaluated in metastatic NSCLC and is discussed regarding the co-occurrence of KRAS mutations [95]. It is a generally accepted biomarker of primary resistance to ICI in KRAS-mutated NSCLC [96]. However, more recent evidence showed poor outcomes among NSCLC with STK11/LKB1 and/or KEAP1 mutations regardless of the treatment received [95,97]. Federico et al. conclude that studies evaluating the impact of STK11 and KEAP1 mutations on outcome in patients treated with ICI or other therapies showed a similar effect, suggesting that this molecular profile should rather be regarded as a prognostic, rather than predictive marker. In a retrospective cohort of 567 localized NSCLC, Pécuchet et al. showed that STK11 mutations could also be assessed according to their potential functional impact. When samples were classified into two groups, exon 1–2 (predicted to lead to an Nterm oncogenic isoform) or exon 3–9 mutated tumors, mutations in exon 1–2 of STK11 delineated an aggressive subgroup with lower OS compared to mutations in exons 3–9 [98]. The results of ongoing trials with adjuvant ICI could evaluate the predictive value of STK11 mutations in localized diseases.

To date, none of these mutations have a validated impact on clinical care, and molecular characterization of localized tumors was not recommended up to now to drive adjuvant treatment. The development of targeted adjuvant treatment opens a new era for molecular testing in localized NSCLC.

### 4.3. TMB

Tumor mutation burden (TMB) has been widely evaluated in metastatic NSCLC as a biomarker for response to ICI therapies [99]. A recent meta-analysis pooled eight different cohorts of five randomized controlled phase III studies (3848 patients with advanced NSCLC) [100]. In TMB-high patients, IO agents were associated with improved ORR and OS when compared with CT, and no benefit was observed in TMB-low cohort. Determination of a threshold for a continuous variable remains difficult. Tumoral heterogeneity cautions against the evaluation of TMB, microenvironmental analysis and immunological signatures from a single locus biopsy [101]. A study of 90 surgically resected NSCLC showed that, considering the median value as a cut-off, patients with high TMB had a trend for lower DFS and significantly lower OS. This was confirmed in multivariate analysis [102]. The authors suggest that high TMB may be involved in resistance to previous adjuvant therapy and could be associated with refractoriness to chemotherapy. However, considering that, similar to metastatic NSCLC, high-TMB patients could benefit from adjuvant chemotherapy, the results of ongoing trials with adjuvant ICI are awaited with great interest.

Concerning SCC, a study based on TCGA data identified that patients with TMB-high tumors had variable outcome depending on the algorithm. The authors thus proposed a gene set enrichment analysis and protein-protein interaction network approach based on TMB high and low, to identify three gene signatures with better discrimination and to propose a prognostic normogram including TMB risk score including TNM [103].

### 4.4. Circulating Tumor DNA (ctDNA)

The detection of ctDNA post-surgery is a validated prognosis factor of high recurrence risk in multiple tumor localizations [104,105]. Different technologies including ultradeep sequencing of cell-free DNA can be used to detect ctDNA as a marker of minimal residual disease (MRD). ctDNA is an interesting marker to monitor recurrence, as positivity precedes radiologic detection. However, detection of ctDNA in the context of MRD remains challenging in lung cancer. The specificity is high, but the sensitivity is low, resulting only in a strong association with relapse in case of positivity and an undetermined status when ctDNA post-surgery is negative. Very small quantities of tumor cell-free DNA with VAF often <0.1% limit the ability of current technologies to ensure the absence of ctDNA. Concerning NSCLC, a recent study using bi-barcoding system and ultradeep sequencing with a limit of detection of the variant allele of 0.01% assayed ctDNA in plasma from 88 patients with resected NSCLC, at baseline, post-surgery, post-ACT (for 64/88) and longitudinal monitoring [106]. Their results showed, as expected, that both post-surgical and post-ACT ctDNA positivity were significantly associated with worse recurrence-free survival. Moreover, in stage II–III patients, the post-surgical ctDNA-positive group benefited from ACT, while ctDNA-negative patients had a low risk of relapse regardless of post-surgery management. This could lead to the use of postsurgical ctDNA status to guide ACT. However, there were only five patients with no ctDNA and no ACT, and larger cohorts are required to consider a clinical trial. Of note, in their study, ctDNA positivity precedes radiological recurrence by a median of 88 days. In another study on 77 patients, preoperative ctDNA positivity was identified as a strong predictor of RFS and OS in localized NSCLC patients undergoing complete resection [107].

### 4.5. Epigenetic

To overcome genomic marker limitations, epigenetic biomarkers have been evaluated. Recent methylation signatures following methylome or microarrays using cancer tissues have been developed with good prognostic value on 143 patients with stage I NSCLC [108]. Tumor expression of microRNA from miR200 family located on chromosome 1 (miR200a,b, 429) was identified as an independent prognostic biomarker in DFS and OS in a series of 176 NSCLC (ADK and SCC) and validated on TCGA data [109]. Validation of epigenetic markers needs to be performed on prospective studies.

### 4.6. Signatures

For all tumor localizations, transcriptomic signatures and multi-omic or pangenomic classification have been used for a decade to describe molecular groups and subtypes of tumors, in parallel to clinical and pathological classification [110,111]. Subgroups of tumors with different prognosis were identified. However, clinical applications are difficult, due to the cost of omic analysis, and to a long analysis and interpretation process. Translation from pangenomic studies to benchmarked biomarkers suitable for clinical routine is a major actual objective. Such a process was conducted in breast cancer to stratify relapse risk using validated and clinically available prognostic signatures. The availability of cheap multi-omic analysis such as the development of 3′RNA-sequencing on FFPE specimens speeds the translation of these technologies from bench to bedside. Concerning NSCLC, 42 prognostic transcriptomic signatures have been published for all stages and all histologic types [112]. However, there is a long-standing debate on the reproducibility, robustness and clinical utility of these expression signatures. In breast cancer series [113], Venet et al. claimed that most signatures are no more significantly associated with survival than randomly generated ones. In lung cancer, Tang et al. performed an important meta-analysis and systematic evaluation of all 42 signatures described [112]. The authors identified 15 NSCLC datasets with more than 50 patients each that were used for the signatures’ prognostic values evaluation. Three different survival association metrics were used to evaluate each prognostic prediction model: hazard ratios (HRs) estimated by the Cox proportional hazards model, the time-dependent receiver-operating characteristics (ROC) curves [8] and the Concordance Index (C.index). The authors showed that 20 out of the 42 published signatures significantly outperformed (*p* < 0.05) random signatures.

The main limitation of the use of lung cancer prognostic signature in clinical practice was the lack of prospective studies. However, a 14 gene signature is awaiting approval and is being tested in prospective trials. The signature was developed on a training cohort of 361 patients enriched in stage I and validated on two external cohorts (433 stage I, 1006 stage I–III). They performed an L1-penalized Cox proportional hazards to select 11/200 genes from previous retrospective data, then an L2-penalized Cox proportional hazards modeling to determine coefficients and produce a continuous score, normalized from 1–100 split in three groups (33rd and 66th centiles) [114]. Woodard et al. showed in a prospective nonrandomized study that this 14 gene signature could identify a subgroup of patients with stage I or IIA with a high risk of recurrence and who would benefit from chemotherapy [115]. A randomized clinical trial is ongoing (NCT01817192) to evaluate ACT in patients with intermediate or high-risk stage I–IIA using the 14 gene signatures. The expected results could change guidelines in the management of early-stage NSCLC.

Of note, in a training cohort of 249 patients with SCC and validation cohort of 234 patients, Bueno et al. identified a prognostic signature that identified a subgroup of stage I and II patients with a high risk of relapse and who could benefit from ACT [116]. This signature could also drive clinical trials evaluating ACT in early stages.

While promising for stratification of relapse risk, molecular signatures could undergo fluctuations related to the tumor heterogeneity. Biswas et al. showed in a cohort of 48 patients with tumors sampled at four different regions that about 1/3 had discordant results using a validated molecular signature [117]. Molecular biomarkers’ prognostic and predictive values are summarized in Table 3.

**Table 3 cancers-14-01400-t003:** Prognostic and predictive values of molecular features in resected lung adenocarcinoma.

Marker	Prognostic Value	Predictive Value
** *EGFR* **	discussed-poor	strong: *EGFR* TKI
** *KRAS* **	discussed-poor	unknown
**MET**	no prognostic value	unknown
**ALK**	discussed-poor	under evaluation ongoing clinical trial
**ROS1/RET**	unknown	unknown
** *TP53* **	discussed-negative prognosis	no predictive value
***KEAP1*/*STK11***	negative prognosis	unknown
**TML**	discussed-negative prognosis	under evaluation ongoing clinical trial
**ctDNA**	negative prognosis	-
**14 gene signatures**	ongoing clinical trial to drive adjuvant chemotherapy in stage I–II	-

### 4.7. Clinical and Histopathological Prognostic Factors in the Era of Peri Operative Immunotherapy

Immunotherapy has revolutionized the management of advanced stages but has also recently made its entry into the modalities of management of localized stages. Indeed, even if the majority of studies are still ongoing, some immunotherapy studies in adjuvant or neoadjuvant settings showed a potential benefit on survival [118].

#### 4.7.1. Adjuvant Situation

Only one study has recently published its results [118]. This is the phase III, ImPower 010, a trial evaluating atezolizumab in patients with NSCLC IB–IIIA undergoing surgery and who received four adjuvant cycles of cisplatin-doublet [119]. The main adjuvant immunotherapy studies (NCT02595944, NCT02273375 and NCT02504372) have completed their recruitment, but no data are available yet. A review has recently reported ongoing adjuvant and neoadjuvant clinical trials [120].

Initial data at median of follow-up showed better median disease-free survival in patients with stage II–IIIA treated by atezolizumab than best support of care (42.3 months versus 35.3 months, HR 0.71; 95% IC (0.64–0.96), *p* = 0.025). In an exploratory analysis, the benefit of adjuvant atezolizumab was particularly pronounced in the PDL 1 ≥ 50% (HR 0.43; 95% IC (0.27–0.68)) and the PDL 1 ≥ 1% (HR 0.66, 95% IC (0.49–0.97)) but was not found in the PDL1 < 1% or PDL 1 (1%–49%); this may suggest that the improvement observed in patients with PDL ≥ 1% is driven by patients with high expression of PDL1. Hence, the determination of PDL1 in resected localized NSLC may play a key role in adjuvant immunotherapy. Interestingly, in the subgroups analyses of all patients at stage II–IIIA: female (HR 0.80 (0.57–1.13)), never smoker (HR 1.13 (0.77–1.67)), EGFR + (0.96 (0.60–1.62)) and ALK + (1.04 (0.38–2.90) seem not to experience significant positive impact with the addition of atezolizumab. However, these results should be interpreted with caution due to the small number of patients in this subgroup [119].

#### 4.7.2. Neoadjuvant Situation

Trials and meta-analysis have been published in the neoadjuvant situation [121,122]. The latest meta-analysis by Ulas et al. included 1066 patients among 19 recent studies. Primary endpoint for most neoadjuvant trials was the major pathological response (MPR) and complete response (pCR) [122]. However, they are good surrogate endpoints for survival [123]. Recent meta-analysis of Ulas et al. found an up to 45% MPR in the mono immunotherapy group, and a higher MPR ranging from 27% to 86% in the immunochemotherapy group. For pCR, it was 0 to 16% in the mono immunotherapy group while it was 9 to 63% in the immunochemotherapy group, suggesting that immunochemotherapy is more effective than immunotherapy alone [118]. However, ICIs are not effective in all patients and may delay or challenge surgery. Biomarkers are actually being analyzed as post hoc studies of clinical trials. Markers linked to response to ICIs in metastatic patients were analyzed in patients with localized cancer. Markers can be grouped as tissue-based markers with PDL-1, tumor mutation load, somatic mutations, immune infiltrate and TCR repertoire; blood-based with circulating immune cells, TCR repertoire and b-TMB or cfDNA or finally host-based with gut microbiome analysis. However, predictive value of common biomarkers such as PDL-1 or TML is variable. In Checkmate 159, MPR was significantly correlated with tumor mutation burden (TMB) before treatment but not with PD-L1 expression [124]. In the Neostar trial, testing nivolumab or nivolumab + ipilimumab as neoadjuvant treatments, histology had no impact on relapse-free survival (RFS) but RFS in non-smokers was lower than in smokers. PDL-1 expression was linked to radiologic responses and MPR; however, there were responses in tumors that were negative for PDL-1 [125]. Changes in the immune infiltrate between biopsy and surgical specimen were higher for patients receiving bitherapy. Nivolumab + ipilimumab induced greater tumor infiltration as compared to nivolumab alone. Sequencing of T-cell receptors (TCRs) was performed in a very small number of patients and suggested increased T-cell richness and clonality in resected tumors compared with pre-therapy samples, although this was not associated with response rates or RFS. At last, exploration of gut microbiome identified gut bacteria species related to MPR, less toxicity and higher T-cell clonality and richness. Circulating DNA may be used to monitor response in patients undergoing neoadjuvant treatment. CtDNA decrease or CtDNA negativity are hallmarks of response [126].

Altogether, and in line with responses in metastatic patients, non-smokers and patients with an oncogene-driven tumor seem not to benefit from ICI in the neoadjuvant setting. In the atezolizumab neoadjuvant trial (LCMC3 (NCT02927301)), patients with EGFR- and ALK-mutated tumors were excluded and neither PDL-1 nor TMB were found to be predictive markers [127].

Data are still needed to refine RFS and OS in patients who have received neoadjuvant immunotherapy prior to surgery.

## 5. Conclusions

Localized primary lung cancer is very heterogeneous in its clinical presentation, histopathology, treatment response and relapse risk after surgery. Lung cancer survival in localized diseases is mainly determined by stage. According to tumor stage, patients will receive adjuvant treatments, but the overall benefit remains low. Systematic assessment of biomarkers to delineate prognosis in patients with lung cancer is not recommended so far; however, the recent implementation of high throughput molecular testing and the development of molecular signature may help stratify relapse risk (Figure 1). In the context of adjuvant-targeted therapies and immunotherapies, cost effectiveness needs to be taken into account and answers may rely on a better stratification of patients. The validation in clinical trial of prognostic signatures is ongoing and could lead to signature-based recommendations to drive adjuvant treatments. Predicting relapse in NSCLC is a major issue; it may finally rely on multiparametrics algorithms, including clinical, histological, molecular data and imaging that need to be developed.

## Figures and Tables

**Figure 1 cancers-14-01400-f001:**
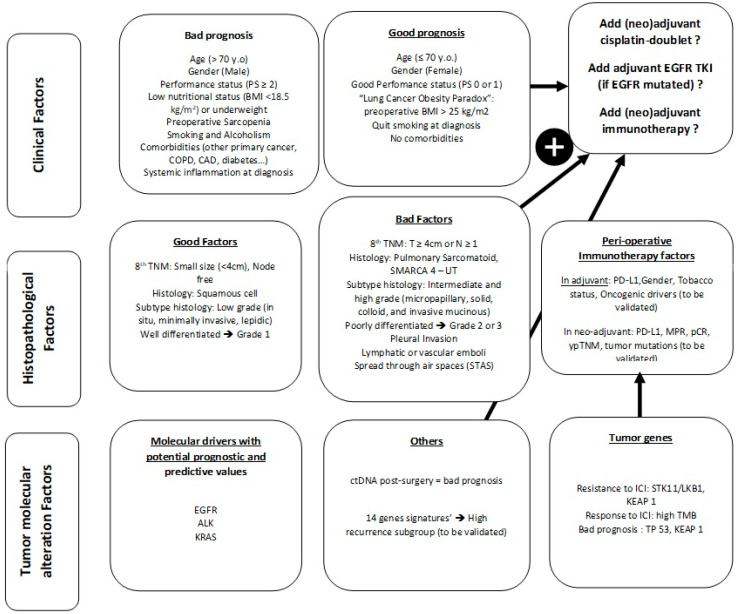
Summary of existing prognostic factors in localized NSCLC.

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
