# Peer review of "Updated Prognostic Factors in Localized NSCLC"

_cancers, 2022, doi:10.3390/cancers14061400_

Round 1

Reviewer 1 Report

The authors of "Updated Prognostic Factors in Localized NSCLC", conceived this review to show the latest prognostic markers that are under investigation or might be applied in a future to better stray patients with early stage lung cancer in order to find candidates that might benefit from adjuvant chemotherapy in resected NSCLC. 

Broadly, the review is well conceived and goes through the main aspects of lung cancer clinicopathological, histological and molecular characteristics. Nevertheless, I have perceived that a substancial part of the bibliography must be updated using more recent studies. 

1. We already have available lung cancer incidence and mortality rates for recent years (Globocan cancer statistics 2020, or "Cancer Statistics 2022" Siegel). There is also in press The 2021 WHO Classification of Lung Tumors: Impact of advances since 2015 published in JTO. 

2. Regarding the TNM classification it would be worth mentioning what are the aims of the 9th Edition of lung cancer TNM classification. A perspective can be found published in Cancers last year.  

3. A major concern given the importance that immunotherapy is gaining in NSCLC, is that a section reviewing more in depth the number of clinical trials that are ongoing up to date to address the relevancy of neoadjuvant or adjuvant immunotherapy in resectable NSCLC, is needed. In this version immunotherapy is slightly mentioned.

4. The role of epigenetic biomarkers in a wide filed of study in NSCLC. I would specify the type of biological sample that has been used in the studies that are mentioned in this review. There are a great number of studies suggesting different types of samples that could be used such as plasma, bronchialveolar lavages...

5. Readers would appreciate a summary chart or figure showing the general conclusions of this review. What are these updated prognostic factors in localized NSCLC. 

Author Response

Response to Reviewer 1 :

The authors of "Updated Prognostic Factors in Localized NSCLC", conceived this review to show the latest prognostic markers that are under investigation or might be applied in a future to better stray patients with early stage lung cancer in order to find candidates that might benefit from adjuvant chemotherapy in resected NSCLC. 

Broadly, the review is well conceived and goes through the main aspects of lung cancer clinicopathological, histological and molecular characteristics.

We thank the reviewer for this positive comment

 Nevertheless, I have perceived that a substancial part of the bibliography must be updated using more recent studies. 

  1. We already have available lung cancer incidence and mortality rates for recent years (Globocan cancer statistics 2020, or "Cancer Statistics 2022" Siegel Cancer Statistics 2022" Siegel https://doi.org/10.3322/caac.21708 USA).
  2. There is also in press The 2021 WHO Classification of Lung Tumors: Impact of advances since 2015 published in JTO.

We thank the reviewer for his remark and updated our manuscript with the suggested recent publications in the introduction. Moreover, changes have been made with recent data in part 2.1, 2.3, and 2.4.

All modifications are highlighted in yellow in the manuscript.

  1. Regarding the TNM classification it would be worth mentioning what are the aims of the 9th Edition of lung cancer TNM classification. A perspective can be found published in Cancers last year.  

A new paragraph has been written to take into account new data from the 9th Edition of lung cancer TNM classification (Section  3.1.4 and 3.2)

  1. A major concern given the importance that immunotherapy is gaining in NSCLC, is that a section reviewing more in depth the number of clinical trials that are ongoing up to date to address the relevancy of neoadjuvant or adjuvant immunotherapy in resectable NSCLC, is needed. In this version immunotherapy is slightly mentioned.

New paragraphs have been written to report recent data on the use of adjuvant and neoadjuvant immunotherapy in localized NSCLC patients. (Section 4.7)

  1. The role of epigenetic biomarkers in a wide filed of study in NSCLC.

I would specify the type of biological sample that has been used in the studies that are mentioned in this review. There are a great number of studies suggesting different types of samples that could be used such as plasma, bronchialveolar lavages...

We added that epigenetic markers mentionned in our review were performed on tumor tissues.

  1. Readers would appreciate a summary chart or figure showing the general conclusions of this review. What are these updated prognostic factors in localized NSCLC. 

A figure has been added (Figure 1) to the manuscript.

Reviewer 2 Report

  1. What is the main question addressed by the research? They should be more up to date with recent studies, it only discusses studies with chemotherapy.
  2. Do you consider the topic original or relevant in the field? Does it address a specific gap in the field? There are many reviews in this field
  3. What specific improvements should the authors consider regarding the methodology? What further controls should be considered? see nº1
  4. Please include any additional comments on the tables and figures. 

*They should refer to prognostic factor data from recent neoadjuvant studies.

*They do not refer to any and these are studies in localized disease, while they do mention studies in advanced disease.

*Table 1- They should explain OR what does it mean. 

Author Response

Response to Reviewer 2

  1. What is the main question addressed by the research? They should be more up to date with recent studies, it only discusses studies with chemotherapy.

The main objectif of our review is to group in a single paper existing clinical, pathological and biological markers of prognosis in resected lung cancer. It may seem less detailed than in a focused review, so as suggested we updated our manuscript with recent studies and included two sections on markers in the field of adjuvant and neoadjuvant immunotherapy in localized NSCLC patients. (Section 4.7)

2. Do you consider the topic original or relevant in the field? Does it address a specific gap in the field? There are many reviews in this field

This review was written to be published in the special issue “Advances in Prognosis and Theranostics of Cancer”. Many changes are ongoing on the management of localized NSCLC with the upcoming of immunotherapy and targeted therapy in this situation. A global evaluation of prognosis will be mandatory to select the good perioperative treatment for patients. So we think that a global review in the field of NSCLC could be of interest for thoracic oncologists.

3. What specific improvements should the authors consider regarding the methodology? What further controls should be considered? see nº1

It is very difficult to have a strict methodology concerning such a wide subject. However, we tried to proceed as follows : relevant published literature was searched for using MEDLINE (PubMed). The following search terms were used: resected OR localized AND non small cell lung cancer AND prognosis.  It brings more than 10.000 results so we then refined the search adding keywords such as « EGFR », « molecular signatures », « WHO classification »…

The exclusion criteria were as follows: 1) case reports, meeting abstracts, letters and expert opinions; and 2) no English translation of the study

We have updated our manuscript with recent publications in 2.1, 2.3, and 2.4.part

All modifications are highlighted in yellow in the manuscript.

4. Please include any additional comments on the tables and figures. 

Figures and Tables are legended, it is not clear to us what additional comments should be added.

*They should refer to prognostic factor data from recent neoadjuvant studies.

*They do not refer to any and these are studies in localized disease, while they do mention studies in advanced disease.

New paragraphs have been written to report recent data on the use of adjuvant and neoadjuvant immunotherapy in localized NSCLC patients. (Section 4.7)

*Table 1- They should explain OR what does it mean. 

It has been done.

Round 2

Reviewer 1 Report

I value that authors incorporated the suggestions made by the reviewers and I feel the review has made substantial improvements. 

Author Response

We thank the reviewer for this positive comment.

Reviewer 2 Report

Some improvement but not enough
For example, in the most deficient part, concerning new developments in the perioperative and neoadjuvant situation, it names only one study and it does it wrong, CM 816 confuses it with one on metastatic disease, reference 124.

Author Response

First we apologize for the reference error, indeed, reference [124] in the text should be :

Forde PM, Chaft JE, Smith KN, Anagnostou V, Cottrell TR, Hellmann MD, Zahurak M, Yang SC, Jones DR, Broderick S, et al. Neoadjuvant PD-1 Blockade in Resectable Lung Cancer N Engl J Med. 2018 May 24;378(21):1976-1986. doi: 10.1056/NEJMoa1716078..

Concerning new developments of neoadjuvant and adjuvant treatments using either ICIs or targeted therapies, we added two paragraphs in the new version: one in the adjuvant, one in the neoadjuvant situation as suggested by both reviewers. In this review, our goal was to provide a general picture of known or discussed prognostic markers in localized lung cancer, form clinical markers to molecular signatures. We focused on prognostic rather than predictive factors of response to treatments. However, both are closely related as adjuvant and neoadjuvant treatments will depend on relapse risk and disease prognostic, and because molecular and cellular markers can themselves impinge upon response to treatment. So adding these paragraphs seemed valuable and we thank reviewers for their proposal.

Reviewer 2 is right, we did not list all ICIs or targeted neoadjuvant and adjuvant trials. This has just been done in a recent review (ref [120]). We chose to discuss selected studies associated with biomarker analysis.

Up to now, no validated marker has emerged from ancillary studies, selection of patients for adjuvant targeted therapy is based, as for metastatic patients, on the presence of the druggable alteration. In a near future tumor, characterization should help better select patients that really benefit from treatments. Concerning ICIs in neoadjuvant and adjuvant settings, most trials are not yet mature to report survival data and the predictive value of biomarkers such as PDL-1, immune infiltrates, TMB or other remains to be validated.

The text was not modified except for reference correction. We hope it will fit your expectations.

Sincerely

The authors

Round 3

Reviewer 2 Report

no more comments, it is ok to me